# Formal consensus to identify clinically important changes in management resulting from the use of cardiovascular magnetic resonance (CMR) in patients who activate the primary percutaneous coronary intervention (PPCI) pathway

Maria Pufulete,[1] Rachel C Brierley,[1] Chiara Bucciarelli-Ducci,[2] John P Greenwood,[3] Stephen Dorman,[2] Richard A Anderson,[4] Jessica Harris,[1] Elisa McAlindon,[5] Chris A Rogers,[1] Barnaby C Reeves[1]

► Prepublication history and additional material is available. To view please visit the journal (http://dx.doi.org/ 10.1136/bmjopen-2016-014627).

For numbered affiliations see end of article.

**Correspondence to**
Dr Maria Pufulete;
maria.pufulete@bristol.ac.uk

## ABSTRACT

**Objective** To define important changes in management arising from the use of cardiovascular magnetic resonance (CMR) in patients who activate the primary percutaneous coronary intervention (PPCI) pathway.

**Design** Formal consensus study using literature review and cardiologist expert opinion to formulate consensus statements and setting up a consensus panel to review the statements (by completing a web-based survey, attending a face-to-face meeting to discuss survey results and modify the survey to reflect group discussion and completing the modified survey to determine which statements were in consensus).

**Participants** Formulation of consensus statements: four cardiologists (two CMR and two interventional) and six non-clinical researchers. Formal consensus: seven cardiologists (two CMR and three interventional, one echocardiography and one heart failure). Forty-nine additional cardiologists completed the modified survey.

**Results** Thirty-seven draft statements describing changes in management following CMR were generated; these were condensed into 12 statements and reviewed through the formal consensus process. Three of 12 statements were classified in consensus in the first survey; these related to the role of CMR in identifying the cause of out-of-hospital cardiac arrest, providing a definitive diagnosis in patients found to have unobstructed arteries on angiography and identifying patients with left ventricular thrombus. Two additional statements were in consensus in the modified survey, relating to the ability of CMR to identify patients who have a poor prognosis after PPCI and assess ischaemia and viability in patients with multivessel disease.

**Conclusion** There was consensus that CMR leads to clinically important changes in management in five subgroups of patients who activate the PPCI pathway.

## INTRODUCTION

There has been rapid uptake of cardiovascular magnetic resonance (CMR) in the UK,

### Strengths and limitations of this study

► We used formal consensus methods to identify potentially important changes in management arising from the use of cardiac magnetic resonance (CMR) in patients who activate the primary percutaneous coronary intervention pathway.

► We identified five patient subgroups in this population in which CMR leads to clinically important changes in management.

► We did not include different stakeholder groups in the consensus panel as the technical wording of the statements would not have been easily understood by non-cardiologists.

with a 44% increase in the number of scans per centre (from 557 to 802) between 2008 and 2010.[1] Despite the rapid increase in the number of scans undertaken, there is limited evidence about the impact of CMR on long-term prognosis or treatment decisions. CMR is increasingly being used for patients who activate the primary percutaneous coronary intervention (PPCI) pathway, but it is not clear how CMR influences clinical management in this population.

We conducted a study to determine the feasibility of setting up a prospective registry using routine data linkage to assess the clinical and cost-effectiveness of CMR in patients who activate the PPCI pathway.[2] A long-term objective of the registry is to compare the incidence of major adverse cardiovascular events (MACE) in patients who do or do not have CMR after the index event. However, given the rare occurrence of such events in these patients, the study would have to accrue over

37 000 subjects to detect a clinically important reduction in the incidence of MACE with adequate power. Therefore, a key objective of the feasibility study was to define a primary composite outcome, acceptable to cardiologists and other stakeholders (eg, clinical commissioners) as representing a clinically important change in management as a result of an eligible patient having had CMR (eg, expected to prevent future MACE) that could be used for the registry in the medium term.

We used a formal consensus method based on the modified nominal group technique[3] to identify important changes in management (and the specific patient subgroups these changes in management relate to) that can be used to define the composite outcome. Formal consensus is a method that combines both research evidence and expert opinion. We used this approach because we knew from a preliminary scoping of the literature that there were few studies that reported the impact of CMR on patient management. There were three components to the consensus process: (1) a literature review to identify evidence about the use of CMR in patients who activate the PPCI pathway, (2) formulation of statements about how CMR changes patient management using research evidence and expert opinion and (3) the formal consensus involving an independent panel of cardiologists using surveys and discussion to reach consensus.

## METHODS
### Literature review and formulation of the consensus statements
An initial working group was convened including two cardiologists with CMR expertise, two interventional cardiologists from sites participating in the feasibility study, one cardiac network director, two methodologists, two statisticians, one health economist and the study manager. We conducted a literature review to search for studies reporting the impact of CMR on prognosis, patient management and risk stratification in the population of patients who activate the PPCI pathway. We searched without restriction by study design or search terms related to outcomes, so that we could determine the full extent of the literature in this area and identify all studies that used CMR in our population. The search was conducted on Medline, Embase, The Cochrane Library, ISI Web of Science (Citations Index and Proceedings) and BIOSIS (see online supplementary appendix 1). The following search terms were used, both as free text and medical subject headings where possible: 'acute coronary syndrome', 'myocardial infarction', 'angioplasty', 'percutaneous coronary intervention', 'cardiovascular magnetic resonance'. We applied no restriction on publication date or language.

Draft statements were generated independently by three of the cardiologists (based on clinical expertise) and by one methodologist (based on evidence from the systematic literature search). Three members of the

working group (study manager, systematic reviewer and cardiologist with CMR expertise) collated the statements, organised them according to patient subgroup and standardised the wording of each statement. A supporting paragraph was drafted for each statement (citing key references from the literature review) to provide background information and put the statement in context. A 1-day meeting was organised for all members of the working group to consider the relevance, format and wording of each statement.

### Survey design
Statements and supporting paragraphs were worded in a consistent manner and collated in the form of a web-based survey (SurveyMonkey, Palo Alto, California, USA) (see online supplementary appendix 2). The survey included an introductory page explaining the purpose and layout of the survey and instructions about how to complete it. Each statement followed with its supporting paragraph containing links to PDF references. Each statement was accompanied by a 9-point Likert scale asking the respondent to indicate whether he/she agreed with the statement or not (with 1 indicating 'completely disagree' and 9 'completely agree'). There was also a free text box for each statement for respondents to comment and justify their score.

### Establishing the expert panel
Clinicians in the working group identified consultant cardiologists with CMR, interventional, echocardiography, electrophysiology and heart failure expertise from across the UK. The cardiologists were invited by email to form an expert consensus panel.

### Expert panel: completion of first survey
The expert panel completed the survey independently. Responses were collated and analysed by members of the working group.

### Expert panel: face-to-face meeting
The expert panel attended a face-to-face meeting, chaired by a non-cardiology clinician experienced in facilitating formal consensus panels. The meeting was also attended by non-clinical members of the working group, whose role was to introduce the study, describe the structure of the formal consensus process, provide study-related information and take minutes of the meeting. The expert panel discussed each statement and anonymised responses to the first survey in turn and agreed on modifications to the survey.

### Expert panel: completion of the modified survey
The survey was modified by members of the working group as agreed in the face-to-face meeting (see online supplementary appendix 3). The expert panel completed the modified survey independently and rated the statements a second time.

### Extension of survey to other UK cardiologists

The survey was extended to other UK cardiologists through the British Cardiovascular Society, which advertised the survey in their monthly newsletter to members (over 2 consecutive months).

### Criteria for consensus

Data for each statement are shown as median and IQR. A median score of ≥7 and IQR of 6–9 were considered to be in agreement or consensus that the change in management described by the statement was clinically important. These statements were used to identify the patient subgroups perceived to benefit from CMR and define the treatment/process outcome that constitutes a definitive management change as a result of having CMR. A median score of ≤3 and IQR of 1–3 were considered to be in consensus that the statement did not constitute a clinically important change in management. Data were analysed using Stata/IC (Version 14).

## RESULTS

### Literature review and formulation of consensus statements

Thirty-seven draft statements were generated by the three cardiologists and one methodologist. The literature search identified a total of 171 studies reporting the use of CMR in patients with acute myocardial infarction (MI) who activated the PPCI pathway. There were no studies that directly compared groups of patients having CMR or not with respect to patient management or clinical outcomes in this population. Statements relating to the same patient subgroup/condition (eg, patients at risk for complications after PPCI who develop left ventricular (LV) thrombus) were condensed into one statement. Participants at the working group combined some statements to avoid repetition and clarified the care pathway without CMR compared with which the additional benefit of CMR was anticipated, for example, alternative imaging modalities such as echocardiography or single-photon emission CT(SPECT). This process resulted in 12 statements (see online supplementary appendix 3) describing changes in management relating to six patient subgroups:

1. patients with poor prognosis or at risk for complications after PPCI (five statements);
2. patients with good prognosis after PPCI who could be discharged earlier and followed up less often (one statement);
3. patients with multivessel disease (MVD) (two statements);
4. patients with unobstructed arteries on angiography (one statement);
5. patients with out-of-hospital cardiac arrest (OHCA) (one statement); and
6. patients with incidental cardiac and extracardiac findings (two statements).

### The expert panel

Nineteen consultant cardiologists were invited to participate in the consensus process. Seven cardiologists (37% of those invited) agreed to participate. Of these, two had CMR, three had interventional, one had echocardiography and one had heart failure expertise. All seven cardiologists completed the first survey independently.

### First survey

There was consensus for 3 of the 12 statements (25%) that the change in management was clinically important (see online supplementary appendix 4): the ability of CMR to identify the cause of OHCA and therefore optimise treatment for the patient; the ability of CMR to provide a definitive diagnosis in patients found to have unobstructed arteries on angiography; and the ability of CMR to identify patients with LV thrombus and initiate treatment with anticoagulation therapy. There was no consensus for any statement that the change in management described was not clinically important.

Consensus was sought on the importance of the change in management, and most respondents commented on this issue. However, some respondents also considered other factors when rating statements, for example, the quality of the supporting evidence, the proportion of patients likely to benefit, the ability of the National Health Service (NHS) to provide a service in line with the statement and whether the cost of CMR justified the perceived benefit.

### Face-to-face meeting

The face-to-face meeting was attended by six of the seven cardiologists and three non-clinical members of the initial working group. As a result of the face-to-face meeting, the survey was modified as follows:

1. The number of statements was reduced from 12 to 10. One statement (relating to patients with MVD) was removed because cardiologists felt that it overlapped significantly with a second statement relating to patients with MVD. Statements relating to incidental cardiac and extracardiac findings were combined into one statement.
2. Respondents were asked to rate five aspects of each statement, organised hierarchically: (a) whether CMR is better than the comparator (eg, echocardiography); (b) whether the information from CMR leads to a change in management; (c) whether the change in management is clinically important (ie, likely to reduce risk of MACE in the long term); (d) whether the change in management is likely to reduce NHS costs in the long term; (e) whether the anticipated benefit was sufficiently large to make CMR cost-effective among the patients in whom it would be indicated (see online supplementary appendix 3). Consensus was based on the distribution of responses to questions (a) to (c) of each statement (ie, median ≥7, IQR 6–9 for clinically important changes in management and median ≤3 and IQR 1–3 for

(a), (b) and (c)). Responses to questions (a) to (c) were used because (a) was constructed to interrogate the respondent's appraisal of the evidence, (b) was constructed to determine whether the respondent believed the NHS delivery changed as a result of CMR and (c) was constructed to determine whether the respondent believed that any change recognised in (b) was clinically important. Discussions in the meeting revealed that consideration of costs and cost-effectiveness for the NHS influenced some panel member's responses to the initial survey. Questions (d) and (e) were added to isolate these aspects of consideration and separate them from the question of change in management, which was the aim of the consensus process.

3. The free text box in which respondents could justify their score was removed. The modified survey is shown in table 1.

## Modified survey

Fifty-four cardiologists (including five of the cardiologists who attended the formal consensus meeting) completed the survey. There was consensus that 5 of the 10 statements (50%) described clinically important changes in management (figure 1). These were statements 3, 5 and 9 (which were in consensus in the first survey) and two additional statements: statement 1, relating to the ability of CMR to identify patients who have a poor prognosis after PPCI, and statement 8, relating to the ability of CMR to assess ischaemia and viability in patients with MVD. There was no consensus that any of the other statements described a change in management that was not clinically important. Examples of comments relating to these five statements from cardiologists who attended the formal consensus meeting are shown in table 2.

## DISCUSSION

We identified five subgroups of acute coronary syndrome (ACS) patients who activate the PPCI pathway for whom there was consensus that CMR changes patient management in a clinically important way: (i) patients who have an OHCA (about 7% of those who activate the PPCI pathway[4]); (ii) patients who have a 'normal' (unobstructed) coronary angiogram (about 10% of those who activate the PPCI pathway[5][6]); (iii) patients who develop LV thrombus (3% overall and 9% in patients with anterior ST-elevation MI (STEMI)[7]; (iv) patients who have MVD (between 40% and 65% of those who undergo PPCI)[8–10]; and (v) patients in whom CMR markers indicate poor prognosis (up to 60% of patients after STEMI).[11][12] There was consensus about the first three subgroups in both rounds of the survey, while consensus was reached about the latter subgroups in the modified survey. These results suggest that CMR benefits a large proportion of patients who activate the PPCI pathway.

## Evidence for the roles for CMR in patients who activate the PPCI pathway

Cardiologists who participated in our research agreed that CMR is superior to echocardiography in establishing a diagnosis in patients who survive an OHCA, which has implications for treatment and prognosis. Despite this view, there are few studies that have reported the role of CMR in managing these patients. An unpublished retrospective case series of 54 OHCA survivors showed that CMR diagnosed the cause (ischaemic or non-ischaemic cardiomyopathy) in 40 (74%).[13]

The cardiologists also agreed that CMR can provide a definitive diagnosis (acute MI, acute myocarditis and cardiomyopathy, especially Takotsubo cardiomyopathy) in patients with ACS and unobstructed coronary arteries. Evidence for the role of CMR in patients with ACS and unobstructed coronary arteries also comes from small retrospective case series. These suggest that CMR provides a definitive diagnosis in 80%–90% of these patients.[14–16] Without access to CMR, the management of these patients is variable in clinical practice, which may have long-term implications. A recent systematic review showed that the overall all-cause mortality in patients presenting with suspected MI and unobstructed coronary arteries was about 5% at 12 months.[17]

The cardiologists agreed that CMR identifies LV thrombus better than echocardiography, which allows more patients to be identified and treated appropriately. Accurate identification of an LV thrombus is important because it often directs subsequent anticoagulation therapy to prevent embolic events. A recent systematic review, published after we conducted the formal consensus study, showed that late gadolinium enhancement CMR is the most accurate modality for detecting LV thrombus, with 88% sensitivity and 99% specificity (compared with routine echocardiography, which had 24%–33% sensitivity and 94%–95% specificity, and contrast echocardiography, which had 23%–61% sensitivity and 96%–99% specificity),[18] although most of the included studies in this review did not use a pathological or surgical gold standard for the detection of LV thrombus.

There was agreement among cardiologists that a CMR-based testing of ischaemia after PPCI would likely optimise the revascularisation strategy for these patients, although they acknowledged that there is no evidence to support the view that a CMR-based revascularisation strategy would improve outcomes for patients with MVD (see table 2). Patients with MVD have a twofold increase in MACE compared with patients with single-vessel disease.[19] There is continuing debate about the benefits of complete versus single-lesion revascularisation during the index PPCI admission.[20] Despite recent evidence from randomised controlled trials showing improved clinical outcomes when complete revascularisation is undertaken at PPCI,[21] current American and European revascularisation guidelines for acute MI recommend

**Table 1** Consensus statements (modified survey)

| | Statement |
|---|---|
| 1 | The following statements relate to the ability of CMR to identify patients who have a poor prognosis after PPCI:<br>a. CMR markers (eg, impaired LV function, large infarct size, microvascular obstruction) better identify patients with a poor prognosis after PPCI than markers based on echocardiography.<br>b. Better identification of patients with a poor prognosis after PPCI allows these patients to be followed up more appropriately and treated more aggressively.<br>c. More appropriate follow-up and more aggressive treatment in these patients are expected to lead to a reduced risk of MACE in the long term. |
| 2 | The following statements relate to the ability of CMR to identify patients who have a good prognosis after PPCI:<br>a. CMR markers (eg, normal LV function, high myocardial salvage, no microvascular obstruction, no residual ischaemia) better identify patients with a good prognosis after PPCI than markers based on echocardiography.<br>b. Better identification of patients with a good prognosis after PPCI allows these patients to be followed up less frequently.<br>c. Less-frequent follow-up in these patients is expected to lead to less NHS resource use in the long term. |
| 3 | The following statements relate to the ability of CMR to identify the causes of OHCA in patients who undergo an emergency angiogram:<br>a. CMR better identifies the cause of OHCA (eg, large myocardial infarction, ARVC, aberrant coronary arteries, HCM) than echocardiography.<br>b. Better identification of the cause of OHCA allows treatment to be optimised for these patients (eg, defibrillator for primary arrhythmia or percutaneous coronary intervention) or their family members (eg, genetic screening and counselling, primary prevention).<br>c. The ability to optimise treatment for these patients or family members is expected to lead to a reduced risk of MACE in the long term. |
| 4 | The following statements relate to the ability of CMR to identify patients with VSD after myocardial infarction:<br>a. CMR identifies the location and characteristics of postinfarct VSD better than echocardiography.<br>b. Better identification of the location and characteristics of postinfarct VSD guides the optimal management of these patients.<br>c. Optimal management of patients with postinfarct VSD is expected to lead to a reduced risk of MACE in the long term. |
| 5 | The following statements relate to the ability of CMR to differentiate myocardial infarction from other diagnoses in patients found to have unobstructed coronary arteries on emergency angiography:<br>a. Unlike echocardiography, CMR can provide a definitive ischaemic diagnosis (eg, myocardial infarction with spontaneous reperfusion or distal embolization) or a non-ischaemic diagnosis (eg, myocarditis, Takotsubo cardiomyopathy, aortic dissection) in patients with unobstructed coronary arteries on angiography.<br>b. A definitive diagnosis results in a patient treatment plan appropriate for that diagnosis.<br>c. A treatment plan appropriate for the diagnosis is expected to lead to a reduced risk of MACE in the long term. |
| 6 | The following statements relate to the ability of CMR to identify patients at high risk for sudden cardiac death after PPCI who would benefit most from an implantable cardiac device:<br>a. CMR identifies PPCI patients who are at high risk for sudden cardiac death better than echocardiography.<br>b. Better identification of PPCI patients at high risk for sudden cardiac death allows optimal patient selection for an implantable cardiac device (ICD or CRT).<br>c. Optimal patient selection for an implantable cardiac device is expected to lead to a reduced risk of MACE in these patients in the long term. |
| 7 | The following statements relate to the ability of CMR to identify patients who would not benefit from CRT after PPCI:<br>a. CMR identifies patients who would not benefit from CRT better than echocardiography.<br>b. The ability to identify patients who would not benefit from CRT would reduce CRT use in patients who do not need it.<br>c. Reducing CRT use in patients who do not need it is expected to lead to reduced risk of MACE in these patients in the long term. |
| 8 | The following statements relate to the ability of CMR to assess ischaemia and viability in patients with multivessel disease:<br>a. CMR assesses ischaemia and viability of the myocardium better than echocardiography.<br>b. Better assessment of ischaemia and viability of the myocardium optimises the revascularisation strategy for patients with multivessel disease and avoids additional diagnostic tests.<br>c. The ability to optimise the revascularisation strategy for patients with multivessel disease is expected to lead to a reduced risk of MACE in the long term. |

**Table 1** Continued

| | Statement |
|---|---|
| 9 | The following statements relate to the ability of CMR to identify patients with postinfarct LV thrombus:<br> a. CMR identifies postinfarct LV thrombus better than transthoracic echocardiography.<br> b. Better detection of postinfarct LV thrombus in PPCI patients allows more affected patients to be treated with anticoagulation therapy.<br> c. Treatment with anticoagulation therapy in patients with postinfarct LV thrombus is expected to lead to a reduced risk of MACE in the long term. |
| 10 | The following statements relate to the ability of CMR to detect incidental cardiac and non-cardiac findings if offered routinely to patients who undergo an emergency angiogram:<br> a. CMR identifies more incidental cardiac/non-cardiac findings than echocardiography.<br> b. Improved detection of potentially significant incidental findings allows affected patients to be investigated further and/or treated.<br> c. Further investigation and treatment are expected to reduce the risk of MACE/increase overall survival in affected patients in the long term. |

ARVC, arrhythmogenic right ventricular cardiomyopathy; CMR, cardiovascular magnetic resonance; CRT, cardiac resynchronization therapy; HCM, hypertrophic cardiomyopathy; ICD, implantable cardioverter defibrillator; LV, left ventricular; MACE, major adverse cardiovascular events; NHS, National Health Service; OHCA, out-of-hospital cardiac arrest; PPCI, primary percutaneous coronary intervention; VSD, ventricular septal defect.

revascularisation only of the infarct-related artery at PPCI in patients with MVD.[22 23]

There was considerable debate about using CMR to identify patients with poor prognosis after PPCI. While there was agreement that CMR parameters of cardiac function (eg, impaired LV function, large infarct size, microvascular obstruction, etc) are useful for risk stratification after STEMI, there was disagreement about whether this would lead to a management change. Some cardiologists felt that there would be no changes to prescribing for secondary prevention since all patients should receive aggressive secondary prevention according to guidelines (see table 2). It was acknowledged, however, that CMR markers are prognostic for outcome (statement 1a in table 1); two recent meta analyses of prognostic studies (conducted after this formal consensus study) showed

an increased risk of MACE, by 13%–15% for every 10% decrease in ejection fraction assessed by CMR,[11] and in patients with microvascular obstruction (MVO) assessed by CMR (OR 2.60, 95% CI 1.68 to 4.02, and OR 4.30, 95% CI 2.19 to 8.43, depending on method of MVO assessment).[24]

### Strengths and limitations

A key strength of this study was the use of a formal consensus approach, based on a systematic search of the literature as well as expert opinion. The nominal group technique is one of the four well-established methodologies for formal consensus (the other three are the Delphi method, RAND/UCLA Appropriateness Method and National Institutes of Health consensus development conference methodology).[2 25] Formal consensus is particularly useful to clarify and standardise practice when relevant and rigorous evidence is lacking. The evidence supporting many of our statements came from small retrospective case series. There were no prospective studies in the literature that reported the impact of CMR on clinical management in our population or whether any changes in management impacted on patient outcomes.

The main features of a formal consensus method are anonymity (statements were rated free from peer-group pressure) and iterative feedback (participants could adjust their initial rating based on the feedback of the group rating). We described the supporting evidence for each statement, identified via the literature review and fed back qualitative (panel members' comments) and quantitative (group median ratings of each statement) information at the face-to-face meeting. We also included a varied group of cardiologists from different specialties in order to encompass diverse perspectives.

The number of panel members in our study was lower than the recommended 8–12 members for a consensus panel.[25] Nevertheless, smaller groups are preferable to

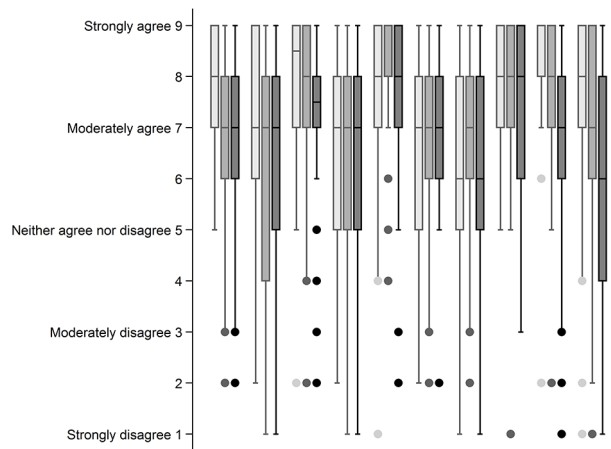

**Figure 1** Median and IQR for the 10 statements in the modified survey (n=54). Boxes represent the median and IQR, and the whiskers represent the range. Dots represent extreme values. Consensus was based on the distribution of responses to a, b and c. Statements 1, 3, 5, 8 and 9 were considered to be in consensus.

**Table 2** Examples of respondents' comments for statements describing changes in management for the three statements (3, 5 and 9) that were in consensus in both the first survey and the modified survey and the two statements (1 and 8) that were in consensus in the modified survey only

| | |
|---|---|
| Statement 3 Patients with OHCA | **Importance of management change**<br>"CMR differentiates causes of cardiomyopathy (IHD vs not). We use it regularly in patients post out-of-hospital cardiac arrest."<br>"I have seen many cases of out-of-hospital cardiac arrest in which the use of CMR has made a diagnosis or significantly altered the diagnosis. Diagnostic refinement in the light of CMR findings also frequently results in changes in drug or device treatment, and often highlights the need for family screening. The improved anatomical, morphological, and tissue characterisation that CMR allows drives the benefit of CMR over echo and allows better identification of the cause of out-of-hospital arrest."<br>**Quality of supporting evidence**<br>"I don't think it has been proved that clinical outcomes are better with a CMR strategy."<br>"CMR is unlikely to alter the immediate management of these patients. However if the cause is unclear after echo and enzymes then CMR is helpful but I am not aware of any studies specifically assessing this." |
| Statement 5 Patients with unobstructed arteries on angiography | **Importance of management change**<br>"Agree. Unobstructed coronaries with elevated troponin need diagnostic resolution and CMR is helpful."<br>"These patients are difficult to manage and often given incorrect diagnosis and especially different theories by different doctors during same admission."<br>"I have direct experience of the benefit of CMR in this area. Therefore, I strongly agree with statement 5. CMR with LGE is especially useful and I have seen many instances when CMR after acute MI with 'normal' coronary arteries has helped diagnosis, differentiating between distal vessel occlusion, LV clot, myocarditis and Takotsubo cardiomyopathy. The results of CMR in this patient group have also directly affected my management of this group of patients including drugs used and length of stay."<br>"MRI is clinically useful in this situation in my experience and occasionally makes a very useful change to management."<br>**Quality of supporting evidence**<br>"Unfortunately again no RCT to indicate benefit of CMR." |
| Statement 9 Patients with postinfarct LV thrombus | **Importance of management change**<br>"Thrombus is poorly assessed by echo with many inconclusive reports in my experience locally."<br>"I strongly agree with this statement as I have experience of many patients in which echo demonstrated no LV thrombus that was subsequently found on CMR. This personal experience correlates with the studies quoted in the text of the statement."<br>**Quality of supporting evidence**<br>"This statement is true but whether this leads to better patient outcomes is uncertain." |
| Statement 1 Patients with poor prognosis after PPCI | **Importance of management change**<br>"All patients who have suffered an acute MI and subsequently undergone PPCI should receive aggressive secondary prevention. Thus, even though CMR can help refine prognosis I do not feel that this additional information would lead to any significant changes to prescribing for secondary prevention. Enhanced confidence in CMR findings (compared with echo) and refinement of prognosis with respect to infarct size, MVO and MSI may help physicians discharge/follow-up patients more appropriately."<br>"All patients should have aggressive secondary prevention."<br>**Quality of supporting evidence**<br>"While I agree with the evidence presented, there is no evidence to support the assertion the CMR findings lead to better outcomes for patients as there have been no trials assessing this."<br>"I don't think it has been proved that CMR compared with echo leads to improved patient outcomes."<br>"While MVO and MSI are markers of prognosis, LV systolic function remains the most important prognostic factor, which can be assessed with echo. We need interventions based on CMR parameters which improve prognosis."<br>**Cost of CMR in relation to perceived benefit**<br>"Agree but not sure current restricted availability and high cost and only modest anticipated change in clinical action justifies wholesale change from echo that also predicts risk well." |

Continued

| Table 2 | Continued | |
|---|---|---|
| Statement 8 Patients with MVD | **Importance of management change** "I'd accept perfusion scanning or DSE as adequate tests for ischaemia and would really only specifically request CMR if there were additional diagnostic questions." "Total revascularisation at one sitting with FFR guidance may render this unnecessary." "Stress CMR in my experience is better than SPECT or stress echo. This is because the improved prognostic and diagnostic accuracy helps physicians manage, with confidence, non-significant coronary disease medically rather than invasively. There are also additional benefits of CMR, for example definition of scar for CRT implant or for VT ablation. It can be helpful to 'archive' this information for latter use if the patient is going to receive a device such as an ICD that may preclude latter CMR scanning." **Quality of supporting evidence** "Evidence shows improved diagnostic accuracy compared with SPECT and DSE, but not in this specific cohort." "The evidence for ischaemia testing in the PPCI era does not really exist. There are no studies comparing MRI in this context with other modalities and definitely no RCT comparing CMR versus another modality." | |

CMR, cardiovascular magnetic resonance; CRT, cardiac resynchronization therapy; DSE, dobutamine stress echocardiography; FFR, fractional flow reserve; ICD, implantable cardioverter defibrillator; IHD, ischaemic heart disease; LGE, late gadolinium enhancement; LV, left ventricular; MI, myocardial infarction; MSI, myocardial salvage index; MVD, multivessel disease; MVO, microvascular obstruction; OHCA, out-of-hospital cardiac arrest; PPCI, primary percutaneous coronary intervention; RCT, randomised controlled trial; SPECT, single-photon emission CT; VT, ventricular tachycardia.

larger ones since, although having more group members increases the reliability of group judgement, large groups reduce the ability to elicit potentially important contributions from every member of the panel.[25] To overcome any criticism of the number of panel members in our study and to prevent the possibility of introducing bias (given the self-selected nature of the panel), we extended the survey to UK cardiologists who did not participate in the formal consensus process. Apart from one statement (statement 1, relating to the ability of CMR to identify patients with poor prognosis after PPCI), the same statements were in consensus when the survey was completed by cardiologists external to the consensus process. This indicates that our process was robust and our survey statements were clear and unambiguous.

A further limitation of our study was that we did not include other stakeholders (eg, general practitioners and patient representatives) in the consensus process.[26] Although we had originally planned to do so, we decided that the specialised nature of the statements would not be easily understood by those without cardiology expertise.

### Conclusion and future research

We have identified the main ways in which cardiologists believe CMR changes clinical management in five subgroups of patients who activate the PPCI pathway. These clinically important changes in management can now be used to design a composite primary outcome for an evaluation of the effectiveness and cost-effectiveness of CMR. This will allow evidence to be obtained more quickly to inform decisions about implementing CMR than would be possible for an evaluation based on MACE.

**Author affiliations**
[1]Clinical Trials and Evaluation Unit, University of Bristol, Bristol, UK
[2]NIHR Bristol Cardiovascular Research Unit, Bristol Heart Institute, University Hospitals Bristol NHS Foundation Trust, Bristol, UK
[3]Multidisciplinary Cardiovascular Research Centre and Leeds Institute of Cardiovascular and Metabolic Medicine, University of Leeds, Leeds, UK
[4]Department of Cardiology, University Hospitals of Wales, Cardiff, UK
[5]Department of Cardiology, New Cross Hospital, Wolverhampton, UK

**Contributors** MP determined the structure of the consensus process, conducted the literature review, formulated statements based on the literature review, was a member of the working group and wrote the manuscript. RB contributed to the consensus process, was a member of the working group, cowrote the manuscript and approved the final manuscript. CB-D conceived the overall feasibility study, contributed to the consensus process, was a member of the working group, revised the manuscript with respect to intellectual content and approved the final manuscript. JPG contributed to the consensus process, was a member of the working group, revised the manuscript with respect to intellectual content and approved the final manuscript. SD contributed to the consensus process, was a member of the working group, revised the manuscript with respect to intellectual content and approved the final manuscript. RAA was a member of the working group, revised the manuscript with respect to intellectual content and approved the final manuscript. JH reviewed the statistical analysis, was a member of the working group, revised the manuscript with respect to intellectual content and approved the final manuscript. EM contributed to the formulation of the consensus statements, revised the manuscript with respect to intellectual content and approved the final manuscript. CAR reviewed the analysis, was a member of the working group and approved the final manuscript. BCR is the chief investigator who conceived the overall feasibility study, including the consensus element, was a member of the working group, revised the manuscript with respect to intellectual content and approved the final manuscript.

**Funding** This study is part of a project funded by the National Institute of Health Research (NIHR) Health Services & Delivery Research (HS&DR 11/2003/58). The British Heart Foundation and NIHR Bristol Biomedical Research Unit for Cardiovascular Disease funded some staff time (CAR, CBD, JH, EM and BCR).

**Disclaimer** The views and opinions expressed are those of the authors and do not necessarily reflect those of the HS&DR programme, NIHR, the UK NHS or the Department of Health.

**Competing interests** JPG has received a research grant from Philips Healthcare. The remaining authors declare that they have no competing interests.

**Ethics approval** NRES Committee South West-Central Bristol.

**Provenance and peer review** Not commissioned; externally peer reviewed.

**Data sharing statement** The literature search based on the published search strategy, details of development of the consensus statements and the survey results can be obtained by contacting the corresponding author (maria.pufulete@bristol.ac.uk).

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
