## [Reviewer comments · BMJ Open]

ARTICLE DETAILS

TITLE (PROVISIONAL)	Formal consensus to identify clinically important changes in management resulting from the use of cardiovascular magnetic resonance (CMR) patients who activate the primary percutaneous coronary intervention (PPCI) pathway
AUTHORS	Pufulete, Maria; Brierley, Rachel; Bucciarelli Ducci, Chiara; Greenwood, John; Dorman, Stephen; Anderson, Richard; Harris, Jessica; MacAlindon, Elisa; rogers, chris; Reeves, Barnaby

VERSION 1 - REVIEW

REVIEWER	Taigang He St George's, University of London, London, UK
REVIEW RETURNED	09-Nov-2016

GENERAL COMMENTS	Cardiovascular magnetic resonance (CMR) is currently playing a pivotal role in clinical cardiology. CMR is increasingly used for a subgroup of ACS patients who activate the primary percutaneous coronary intervention pathway, but it seems unclear how CMR influences clinical management in this population. This consensus study is well designed to answer this question and the message is simple, clear but could be very useful to the society. The study is well presented in general with minor issues. I would therefore recommend for a publication with minor revision. 1. Page 1. The title needs to be reworded The objective is to determine important changes in patient management arising from the use of cardiovascular magnetic resonance (CMR) imaging; but the original title can be simplified as "consensus to identify management change resulting from CMR patients...". This is incomplete or confusing. It could be '... use of CMR in patients...' from my understanding. 2. Page 5. Strengths and limitations of this study Please consider remove the last sentence. This is not really a limitation to the current study as no single study can address all patient groups that may benefit from CMR. 3. Page 6. consider rewriting: '...for patients with acute coronary syndrome (ACS) who activate the primary percutaneous coronary intervention (PPCI) pathway...'. To note, patients but not ACS activate PPCI pathway. 4.. The term cardiovascular magnetic resonance (CMR) is the convention in the society. Please check the whole manuscript as CMR was abbreviated for both 'cardiovascular magnetic resonance'
--

	and 'cardiovascular magnetic resonance imaging'.
REVIEWER	Eike Nagel Institute for Experimental and Translational Cardiovascular Imaging DZHK Centre for Cardiovascular Imaging Goethe University Frankfurt am Main
REVIEW RETURNED	20-Feb-2017

GENERAL COMMENTS	This is an interesting concept on generating consensus on patients “activating the PPCI pathway”. Unfortunately, the paper is neither as study, nor describing a novel method, nor a proper consensus statement. 1.) The aims and goals of the “study” are not well described:  • After reading the title and the abstract I was still unclear what this is about. What patients exactly are looked at? What is a “formal consensus study”. The paper is not a study, it is a consensus statement generated by an interactive process of (random) participants. • The reason for the study was “to determine the feasibility of setting up a prospective registry”, later it is stated, that “the key objective ... was to define a primary composite outcome”. • Neither of the two reasons is addressed in the “study” 2.) The authors need to decide, whether this is a study assessing the recognition of CMR in patients after PPCI within the larger cardiology community of the UK or whether this is a consensus statement prepared by iterative majority consensus vote of some UK cardiologists. The method used has been described before and this should be introduced in the introduction and discussed in the methods. (It partially is). It would also be good to compare this approach to the ACC/AHA approach for Appropriateness Criteria. 3.) There is redundancy of the addenda and the main paper. It would help readability of the main paper if the statements receiving final support would be ordered as 1 – X and the ones without support X – 12. The process, the first version of statements and the results from the first round could then be placed solely in the addendum. 4.) Some of the evidence is questionable and needs to be used with more care. E.g. the meta-analysis on CMR being superior to echo for the detection of thrombus includes several papers, which use CMR LGE as the reference standard. Such evidence should not be used. Obtaining consensus and defining the utility of CMR has merits and the paper adds knowledge. I would suggest to focus on the consensus, do not call it a study and place the pathway for obtaining the consensus in the addendum
--

VERSION 1 – AUTHOR RESPONSE

Reviewer 1

1. Page 1. The title needs to be reworded. The objective is to determine important changes in patient management arising from the use of cardiovascular magnetic resonance (CMR) imaging; but the

original title can be simplified as "consensus to identify management change resulting from CMR patients...". This is incomplete or confusing. It could be '... use of CMR in patients...' from my understanding.

We have changed the title to "Formal consensus to identify clinically important changes in management resulting from the use of cardiovascular magnetic resonance imaging (CMR) patients who activate the primary percutaneous coronary intervention (PPCI) pathway".

2. Page 5. Strengths and limitations of this study. Please consider remove the last sentence. This is not really a limitation to the current study as no single study can address all patient groups that may benefit from CMR.

This sentence has been removed.

3. Page 6. consider rewriting: '...for patients with acute coronary syndrome (ACS) who activate the primary percutaneous coronary intervention (PPCI) pathway...'. To note, patients but not ACS activate PPCI pathway.

We have removed the term "acute coronary syndrome (ACS)" to clarify the sentence.

4. The term cardiovascular magnetic resonance (CMR) is the convention in the society. Please check the whole manuscript as CMR was abbreviated for both 'cardiovascular magnetic resonance' and 'cardiovascular magnetic resonance imaging'.

We have amended all relevant text to state cardiovascular magnetic resonance (CMR) or CMR.

Reviewer 2

This is an interesting concept on generating consensus on patients "activating the PPCI pathway". Unfortunately, the paper is neither as study, nor describing a novel method, nor a proper consensus statement.

We disagree with this comment. Our research represents a study; the Nominal Group Technique is one of the four well-established methodologies for formal consensus (organizing subjective judgments and synthesising them with the available evidence); the other three are the Delphi method, RAND/UCLA Appropriateness Method (RAM), and National Institutes of Health (NIH) consensus development conference methodology. Our study is novel in that it is the first to attempt to identify potentially important changes in management arising from the use of cardiac magnetic resonance (CMR) in a specific patient group. We are not sure what the reviewer means by a "proper consensus statement" but we met our study aim of identifying important changes in management (and the specific patient subgroups these changes in management relate to) resulting from the use of CMR in a specified patient population. We identified five subgroups of ACS patients who activate the PPCI pathway for whom there was consensus that CMR changes patient management in a clinically important way (i.e. expected to prevent adverse clinical outcomes in the long term).

1. The aims and goals of the "study" are not well described:

- After reading the title and the abstract I was still unclear what this is about. What patients exactly are looked at?

We have made it clear in the title and the abstract that our study population is "patients who activate the primary percutaneous coronary intervention (PPCI) pathway".

- What is a “formal consensus study”.

See response to general comments from Reviewer 2 above.

- The paper is not a study, it is a consensus statement generated by an interactive process of (random) participants.

See response to general comments from Reviewer 2 above.

- The reason for the study was “to determine the feasibility of setting up a prospective registry”, later it is stated, that “the key objective ... was to define a primary composite outcome”. Neither of the two reasons is addressed in the “study”.

We have explained clearly in the introduction that a main objective of the feasibility study was to define a primary composite outcome based on clinically important changes in management, and that we used a formal consensus method to identify these clinically important changes in management. We have made it clear in the conclusions and future research section that the “clinically important changes in management will now be used to design a composite primary outcome for an evaluation of the effectiveness and cost-effectiveness of CMR”.

2. The authors need to decide, whether this is a study assessing the recognition of CMR in patients after PPCI within the larger cardiology community of the UK or whether this is a consensus statement prepared by iterative majority consensus vote of some UK cardiologists. The method used has been described before and this should be introduced in the introduction and discussed in the methods. (It partially is). It would also be good to compare this approach to the ACC/AHA approach for Appropriateness Criteria.

The study is a consensus study to identify clinically important changes in management resulting from CMR in a specific patient population. The formal consensus method based on the modified nominal group technique used in our study is presented and referenced in the introduction (page 6, second line paragraph 3). The separate components of the nominal group technique are fully explained in the introduction (page 6 & 7, end of paragraph 3). We have added a statement about the different consensus development methods (page 24, paragraph 1), but it is beyond the scope of this paper to compare the different approaches.

3. There is redundancy of the addenda and the main paper. It would help readability of the main paper if the statements receiving final support would be ordered as 1 – X and the ones without support X – 12. The process, the first version of statements and the results from the first round could then be placed solely in the addendum.

We have removed the results of the first survey from the main paper and placed these in appendices to improve readability. However, we would prefer not to change the order of the statements (i.e. place those that received support first) as suggested by the reviewer; the number order relates specifically to how the statements were developed and refined from the beginning to the end of the consensus process. We have also chosen to keep comments relating to the statements from the cardiologists who participated in the formal consensus meeting in the paper. Although these comments were part of the first round of the survey and the face-to-face meeting, they add important contextual information that will help readers interpret the results.

4. Some of the evidence is questionable and needs to be used with more care. E.g. the meta-analysis on CMR being superior to echo for the detection of thrombus includes several papers, which use

CMR LGE as the reference standard. Such evidence should not be used.

We have included a sentence stating that “most of the included studies in this review did not use a pathological or surgical gold standard for the detection of LV thrombus” (see page 23, paragraph 2).

Obtaining consensus and defining the utility of CMR has merits and the paper adds knowledge. I would suggest to focus on the consensus, do not call it a study and place the pathway for obtaining the consensus in the addendum

We have taken all comments on board, except for the comment about referring to this paper as a study (see response to general comments from Reviewer 2 above).

Please let us know if you need further information or clarification. I look forward to hearing from you.

VERSION 2 – REVIEW

REVIEWER	Taigang He St George's, University of London
REVIEW RETURNED	02-Apr-2017

GENERAL COMMENTS	The authors have responded to reviewers comments. I feel the manuscript is much improved after the revision and I would recommend for publication in your journal.
--